# eIF4A, a target of siRNA derived from rice stripe virus, negatively regulates antiviral autophagy by interacting with ATG5 in *Nicotiana benthamiana*

Xiangxiang Zhang[1,2], Yueyan Yin[1,2], Yunhe Su[1], Zhaoxing Jia[1], Liangliang Jiang[1], Yuwen Lu[1], Hongying Zheng[1], Jiejun Peng[1], Shaofei Rao[1], Guanwei Wu[1], Jianping Chen[1,2,3] *, Fei Yan[1] *

1 State Key Laboratory for Managing Biotic and Chemical Threats to the Quality and Safety of Agro-products, Institute of Plant Virology, Ningbo University, Ningbo, China, 2 Plant Protection College, Yunnan Agricultural University, Kunming, China, 3 Key Laboratory of Biotechnology in Plant Protection of Ministry of Agriculture and Zhejiang Province, Institute of Virology and Biotechnology, Zhejiang Academy of Agricultural Sciences, Hangzhou, China

* jianpingchen@nbu.edu.cn (JC); yanfei@nbu.edu.cn (FY)

**Data Availability Statement:** All relevant data are within the manuscript and its Supporting Information files.

## Abstract

Autophagy is induced by viral infection and has antiviral functions in plants, but the underlying mechanism is poorly understood. We previously identified a viral small interfering RNA (vsiRNA) derived from rice stripe virus (RSV) RNA4 that contributes to the leaf-twisting and stunting symptoms caused by this virus by targeting the host eukaryotic translation initiation factor 4A (eIF4A) mRNA for silencing. In addition, autophagy plays antiviral roles by degrading RSV p3 protein, a suppressor of RNA silencing. Here, we demonstrate that eIF4A acts as a negative regulator of autophagy in *Nicotiana benthamiana*. Silencing of NbeIF4A activated autophagy and inhibited RSV infection by facilitating autophagic degradation of p3. Further analysis showed that NbeIF4A interacts with NbATG5 and interferes with its interaction with ATG12. Overexpression of NbeIF4A suppressed NbATG5-activated autophagy. Moreover, expression of vsiRNA-4A, which targets NbeIF4A mRNA for cleavage, induced autophagy by silencing NbeIF4A. Finally, we demonstrate that eIF4A from rice, the natural host of RSV, also interacts with OsATG5 and suppresses OsATG5-activated autophagy, pointing to the conserved function of eIF4A as a negative regulator of antiviral autophagy. Taken together, these results reveal that eIF4A negatively regulates antiviral autophagy by interacting with ATG5 and that its mRNA is recognized by a virus-derived siRNA, resulting in its silencing, which induces autophagy against viral infection.

## Author summary

Autophagy is induced by viral infection and has antiviral functions in plants, but the underlying mechanism is poorly understood. Here we demonstrate that eIF4A is a negative regulator of autophagy in *N. benthamiana* and rice that acts by inhibiting the function

**Funding:** This work was supported by the National Natural Science Foundation of China awarded to F. Y. (31772239) and to S.R. (31901849), the National Transgenic Science and Technology Program awarded to F.Y. (2016ZX08001-002), the Natural Science Foundation of Ningbo to S.R. (2019A610408), and the K.C. Wong Magna Fund of Ningbo University to F.Y. The funders had no role in study design, data collection and analysis, decision to publish, or preparation of the manuscript.

**Competing interests:** The authors have declared that no competing interests exist.

of ATG5, a key component of autophagy. We previously reported that *NbeIF4A* transcripts are targeted for cleavage by rice stripe virus (RSV)-derived siRNA and that autophagy plays an antiviral role by degrading RSV p3 protein, a suppressor of RNA silencing. Together, our findings reveal a novel mechanism in which a negative regulator of antiviral autophagy recognizes a virus-derived siRNA and sacrifices itself to induce autophagy, which inhibits viral infection.

## Introduction

Eukaryotic initiation factor 4A (eIF4A), a member of the DEAD-box RNA helicase protein family, is thought to use the energy from ATP hydrolysis to unwind the mRNA structure and to prepare mRNA templates for ribosome recruitment during translation initiation, together with other components [1]. The *Arabidopsis thaliana* genome encodes two isoforms of eIF4A, one of which (eIF4A-1) is required for the coordination between cell cycle progression and cell size [2]. *eif4a1* mutants display slow growth, reduced lateral root formation, delayed flowering, and abnormal ovule development [2]. A T-DNA mutation of *eIF4A* confers dwarfing in *Brachypodium distachyon* in a dose-dependent manner [3]. Therefore, eIF4A is essential for plant growth and development [2,3]. We previously determined that the mRNA of *eIF4A* from *N. benthamiana* (Nb*eIF4A*) is targeted by a viral small interfering RNA (vsiRNA-4A) produced from the genomic RNA4 fragment of rice stripe virus (RSV) by mRNA cleavage [4]. Silencing of Nb*eIF4A* caused dwarfing in plants, which is consistent with findings for *B. distachyon* [3,4]. However, the role of eIF4A in plant–virus interactions remains unclear.

Autophagy is an evolutionarily conserved mechanism that employs double-membrane vesicular autophagosomes to enclose and deliver cytoplasmic material for vacuolar or lysosomal degradation and recycling [5]. Autophagy-related genes (ATGs) encode key factors in this process. In plants, autophagy plays essential roles in development, reproduction, metabolism, senescence, and tolerance to abiotic and biotic stress [6,7]. Autophagy is induced in response to viral infection; increasing evidence suggests that autophagy participates in antiviral responses during plant–virus interactions [7–10]. For example, autophagy is induced during the resistance response to tobacco mosaic virus (TMV), and plants lacking ATG activity exhibit enhanced virus accumulation [11].

Several viral proteins were recently shown to be targeted by autophagy for degradation, thus limiting viral infection. For instance, the autophagy cargo receptor NEIGHBOR OF BREAST CANCER 1 (NBR1) targets the capsid protein (CP) of cauliflower mosaic virus (CaMV) and the helper component proteinase of turnip mosaic virus (TuMV) for autophagic degradation and suppresses viral accumulation [12,13]. The TuMV protein NIb is targeted by Beclin1 (also called ATG6), a core autophagy component that mediates NIb degradation [14]. The virulence factor βC1 of cotton leaf curl Multan virus (CLCuMuV) interacts with the key autophagy protein ATG8 and is degraded by the autophagic machinery [15]. We previously demonstrated that p3, a viral suppressor of RNA silencing (VSR) from rice stripe virus (RSV), interacts with NbP3IP, a cargo receptor from *Nicotiana benthamiana*, and is delivered to autophagic vesicles for degradation [16]. In addition, protein 2b from cucumber mosaic virus (CMV) is thought to be targeted for degradation by autophagy through the calmodulin-like protein rgs-CaM [17].

Although the antiviral function of autophagy has been well established in plants, how autophagy is induced by viral infection is not well understood [13–15,18]. Ismayil et al. recently reported that CLCuMuV βC1 protein interacts with the negative autophagic

regulators glyceraldehyde-3-phosphate dehydrogenases (GAPCs) to induce autophagy in plants [19,20]. We previously identified RNA4, a small interfering RNA derived from RSV that targets and silences the host mRNA encoding eIF4A, thus contributing to the leaf-twisting and stunting symptoms observed during RSV infection [4]. Here we discovered that eIF4A acts as a negative regulator of antiviral autophagy in *N. benthamiana*, thus defining a mechanism whereby a negative regulator of antiviral autophagy recognizes a virus-derived siRNA and sacrifices itself to induce autophagy against viral infection.

## Results

### Silencing of Nb*eIF4A* inhibits RSV infection in *Nicotiana benthamiana*

We previously determined that the mRNA of *eIF4A* from *N. benthamiana* is targeted by a viral small interfering RNA (vsiRNA-4A) produced from the genomic RNA4 fragment of RSV by mRNA cleavage [4]. Here, we tested whether the downregulation of *eIF4A* would have any effects on RSV infection. We silenced *N. benthamiana eIF4A* (Nb*eIF4A*) in plants with the *Tobacco rattle virus* (TRV)-based virus-induced gene silencing (VIGS) system by inserting a DNA fragment specific for Nb*eIF4A* into RNA2 of the VIGS vector TRV to generate the TRV:4A construct. We monitored the progression of symptoms linked to RSV infection in TRV:4A-infected and control plants. At 15 days post infiltration (dpi), plants infected with TRV:4A alone showed leaf-twisting, whereas plants infected with the empty VIGS vector TRV:00 displayed no obvious morphological changes (Fig 1A). RT-qPCR analysis revealed a reduction in Nb*eIF4A* transcript levels of approximately 60% in TRV:4A-infected plants relative to the TRV:00 controls (Fig 1B). These results are consistent with a previous report [4].

Next, we inoculated Nb*eIF4A*-silenced (TRV:4A-infected) or control (TRV:00-infected) plants with RSV. At 20 dpi with RSV, typical RSV symptoms, such as stunting and leaf-twisting with yellow mosaicism, appeared in all TRV:00 control plants (Fig 1C and 1D). By contrast, only 20% of RSV-inoculated Nb*eIF4A*-silenced plants showed yellowing and mosaicism in leaves, which was less pronounced than that of the control (Figs 1C, 1D and S1). RT-PCR confirmed the systemic infection by RSV (S2 Fig). Both virulence and systemic infection by RSV were reduced in Nb*eIF4A*-silenced vs. control plants. Immunoblot analysis revealed lower accumulation of RSV CP in RSV-infected Nb*eIF4A*-silenced plants relative to TRV:00 control plants. These results demonstrate that silencing of Nb*eIF4A* inhibits RSV infection of *N. benthamiana* plants (Figs 1E and S2).

### Silencing of Nb*eIF4A* activates autophagy

In our recent report, autophagy is demonstrated to play a key role in plant defense against RSV infection [16]. To determine whether autophagy pathway helps prevent RSV infection in Nb*eIF4A*-silenced plants, we investigated the expression of genes in autophagy upon Nb*eIF4A* silencing. Results showed that genes functioning in autophagy, such as Nb*ATG2*, Nb*ATG3*, Nb*ATG5*, Nb*ATG6*, Nb*ATG7*, and *PHOSPHOINOSITIDE 3-KINASE* (Nb*P13K*), were significantly upregulated in Nb*eIF4A*-silenced plants (S3 Fig), suggesting that autophagy may be activated in these plants.

To directly test this hypothesis, we tagged *N. benthamiana* ATG8f with cyan fluorescent protein (CFP) at its N terminus (CFP-*Nb*ATG8f) to monitor autophagic activity in Nb*eIF4A*-silenced cells, as previously described [20]. We observed a diffuse fluorescent signal in plants infected with the empty vector TRV:00, whereas silencing of Nb*eIF4A* resulted in more fluorescent foci, indicative of activated autophagy in Nb*eIF4A*-silenced cells (Fig 2A and 2B). We independently validated these results using the stain monodansylcadaverine (MDC), which accumulates in acidic autophagic vacuoles (Fig 2C and 2D). Finally, we observed multiple

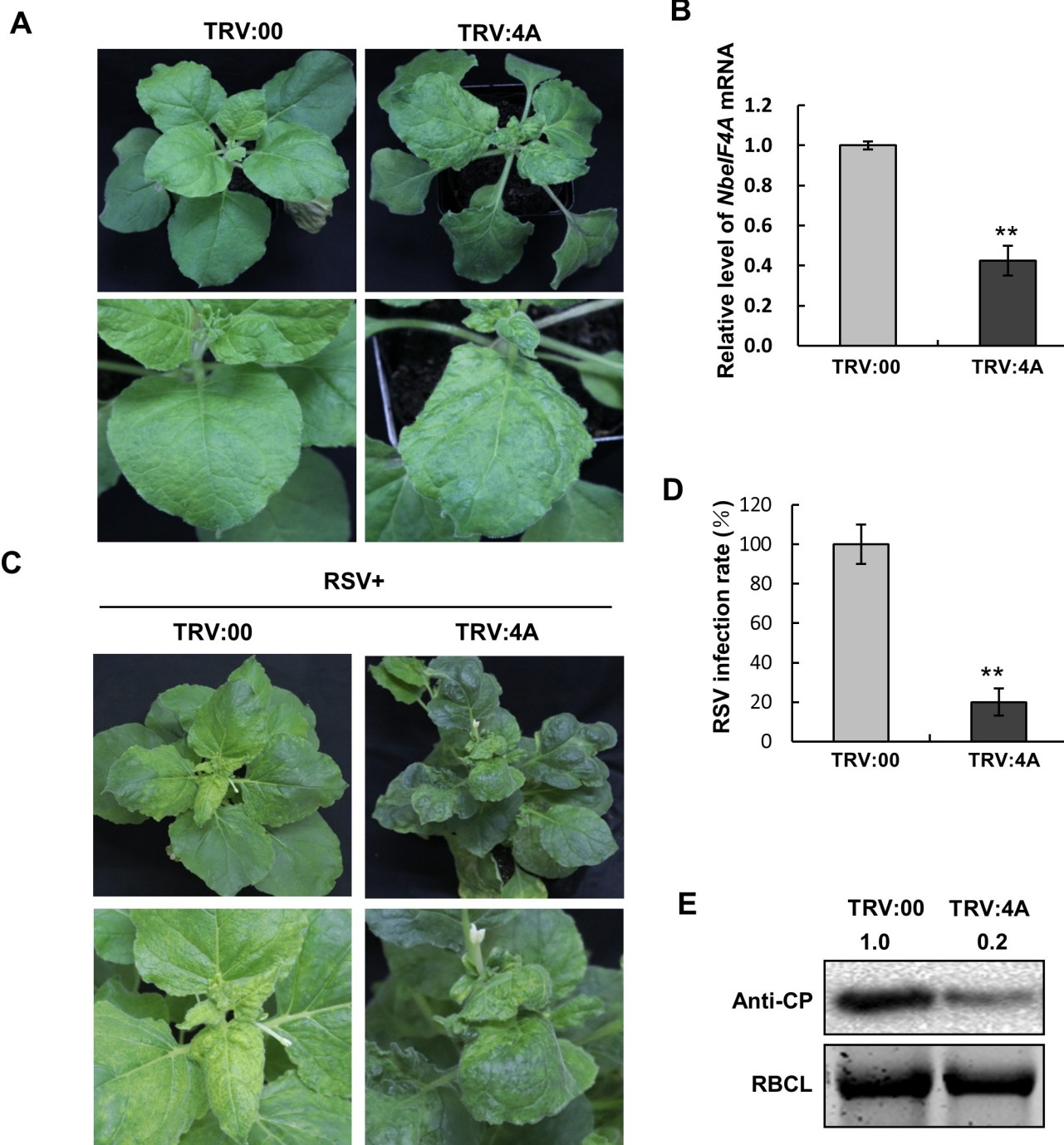

**Fig 1. Silencing of Nb*eIF4A* Prevents RSV Infection in *N. benthamiana*.** (**A**) Leaf-twisting phenotype of Nb*eIF4A*-silenced *N. benthamiana* plants via TRV-induced gene silencing at 15 dpi. Bottom panel: close-up view of a leaf. (**B**) RT-qPCR analysis of Nb*eIF4A* transcript levels in TRV:00- and TRV:4A-infected *N. benthamiana* plants. *NbActin* served as an internal control. Bars indicate standard error from three individual experiments. (**C**) Strong typical symptoms of RSV infection in control plants (TRV:00) and mild symptoms in Nb*eIF4A*-silenced plants (TRV:4A) at 20 dpi with RSV. Bottom panel: close-up view of a leaf. (**D**) RSV infection rate in Nb*eIF4A*-silenced plants. Bars indicate standard error from three individual experiments. Twenty plants were used for each experiment. (**E**) Immunoblot analysis of RSV CP accumulation in control and *NbeIF4A*-silenced plants. An anti-RSV CP antibody was used. Asterisks indicate significant differences by Student's *t*-test compared to the control (\*\*, $p < 0.01$). Band intensity was analyzed by ImageJ.

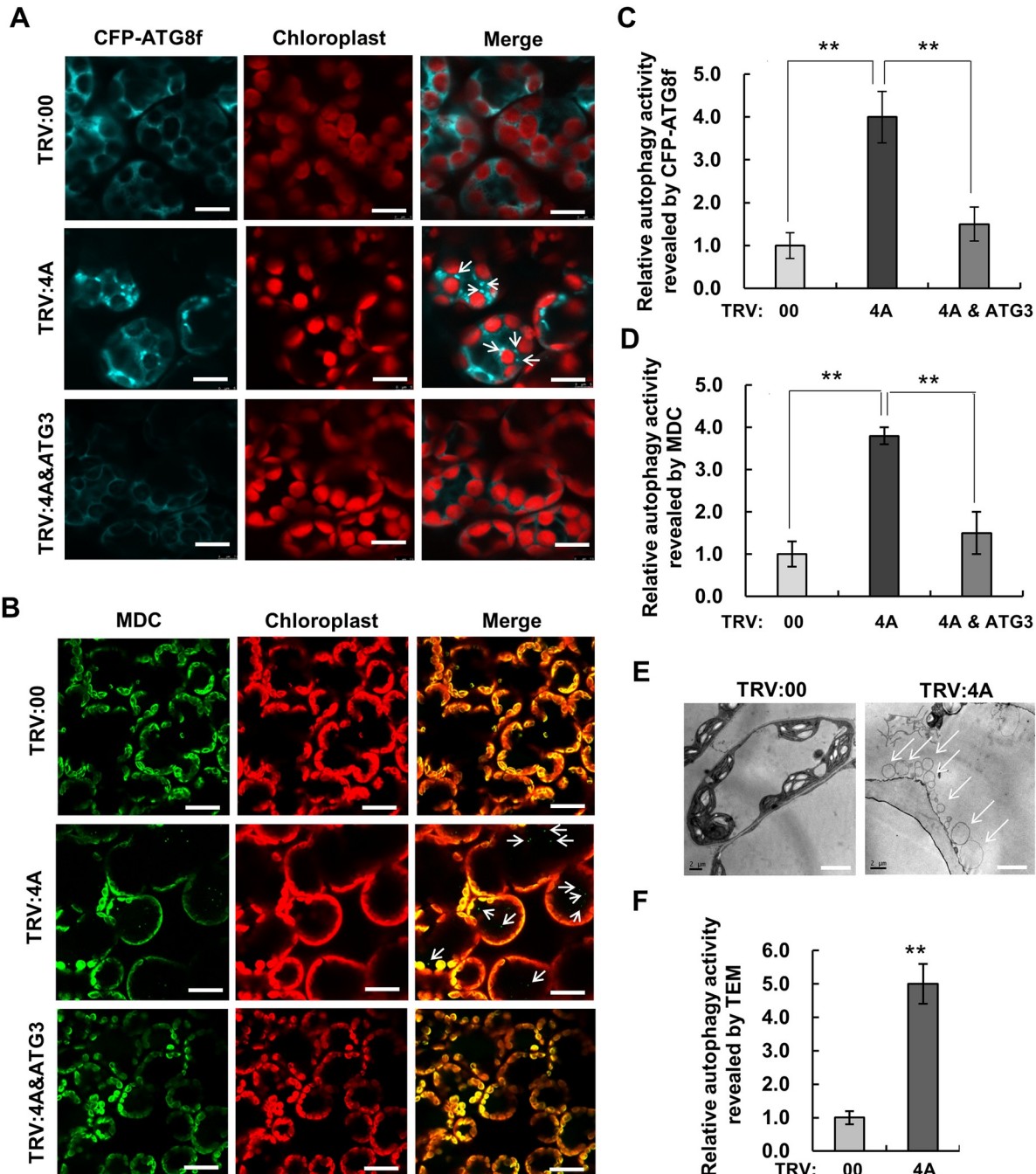

**Fig 2. Silencing of Nb*eIF4A* Activates Autophagy. (A, B)** Representative confocal images of dynamic autophagic activity, as revealed by the specific autophagy marker CFP-NbATG8f **(A)** or MDC staining **(B)** in control (TRV:00), Nb*eIF4A*-silenced (TRV:4A) plants, and Nb*ATG3* and Nb*eIF4A* (TRV:4A&ATG3) co-silenced plants. Autophagosomes and autophagic bodies labeled by CFP-NbATG8f **(A)** or MDC staining **(B)** in mesophyll cells are indicated by white arrows. Cyan: CFP-*Nb*ATG8f-labeled punctate autophagosomes and autophagic bodies; green: MDC-labeled punctate autophagosomes and autophagic bodies; red: chlorophyll autofluorescence. Numerous autophagosomes and autophagic bodies were observed inTRV:4A-infected cells, but fewer were observed in TRV:00-or TRV:4A&ATG3-infected cells. Bars, 10 μm. **(C, D)** Relative autophagic activity in silenced cells, normalized to control TRV:00-infected cells, which was set to 1.0. Quantification of CFP-ATG8f-labeled **(C)** or MDC staining **(D)** autophagic foci per cell was performed by counting the autophagic bodies as a proxy for autophagic activity. Over 150 cells per treatment were used for quantification. Bars indicate standard error from three individual experiments. Asterisks indicate significant differences by Student's *t*-test compared to the control (\*\*, $p<0.01$). **(E)** TEM analysis of TRV: 00- and TRV:4A-inoculated *N. benthamiana* cells. White arrows indicate autophagosomes. **(F)** Relative autophagic activity in Nb*eIF4A*-silenced plants revealed by TEM analysis, normalized to that of TRV:00 control plants, which was set to 1.0. Approximately 20 cells were used to quantify autophagic structures in each treatment. Experiments were repeated three times. Asterisks indicate significant differences by Student's *t*-test compared to mock inoculation (\*\*, $p<0.01$).

autophagic structures in Nb*eIF4A*-silenced cells by transmission electron microscopy (TEM) and far fewer in TRV:00 control plants (Fig 2E and 2F). These results support our hypothesis that autophagy is activated in Nb*eIF4A*-silenced cells.

To exclude the possibility that silencing of Nb*eIF4A* affected general translation in cells, we examined whether protein translation was affected by silencing of Nb*eIF4A*. For this analysis, we developed an RNAi construct expressing hairpin RNA of Nb*eIF4A* (4A-hairpin) to silence Nb*eIF4A*. As a negative control, we generated a construct expressing hairpin RNA of β-glucuronidase (GUS-hairpin). As a positive control, we used a construct expressing hairpin RNA of Nb*eIF6A* (6A-hairpin), which plays key roles in protein translation [21]. These three constructs were individually infiltrated into a single leaf of a plant with construct of 35S-driven GFP via Agrobacterium-mediated infiltration. At 3 dpi, in zones expressing 6A-hairpin where the expression of Nb*eIF6A* was silenced, GFP accumulated to a lower level than that in zones expressing GUS-hairpin or 4A-hairpin. By contrast, there was no significant difference in GFP accumulation between zones expressing GUS-hairpin and 4A-hairpin (S4 Fig). These results indicate that protein translation was not significantly affected by silencing of Nb*eIF4A*, suggesting that the activated autophagy in Nb*eIF4A*-silenced cells was not due to the suppression of protein translation. Meanwhile, results cannot rule out that NbeIF4A has any role in RSV translation.

To further characterize the activation of autophagy in Nb*eIF4A*-silenced plants, we also silenced Nb*ATG3*, which encodes a key component of autophagy. Using CFP-*Nb*ATG8f as a reporter of autophagic activity, we established that plants co-silenced for both Nb*eIF4A* and Nb*ATG3* accumulated fewer fluorescent foci (representing CFP-*Nb*ATG8f-positive autophagosomes) compared to Nb*eIF4A*-silenced plants, supporting our hypothesis that silencing of Nb*eIF4A* activates autophagy (Figs 2A, 2B and S5).

NIb, a viral RNA-dependent RNA polymerase (RdRp) of TuMV, interacts with Beclin (ATG6) and is degraded by autophagy [14]. We reasoned that if silencing of *NbeIF4A* induced autophagy, silenced plants should be more resistant to TuMV infection than the control. To test this hypothesis, we inoculated Nb*eIF4A*-silenced plants and TRV:00 control plants with TuMV by Agrobacterium (*Agrobacterium tumefaciens*)-mediated infiltration and monitored the progression of infection with the GFP reporter. At 3 dpi, Nb*eIF4A*-silenced plants accumulated fewer fluorescent foci on locally infiltrated leaves compared to control plants (S6A Fig).

Consistent with this observation, TuMV accumulated at a lower level in local silenced leaves than the control, as evidenced by the green fluorescent protein (GFP) signal derived from the GFP-tagged TuMV [22] (S6B Fig). At 6 dpi, GFP fluorescence had spread to newly emerging leaves of control plants, indicating the complete systemic infection of TuMV (S6A Fig). By contrast, GFP fluorescence was very weak in the new leaves of Nb*eIF4A*-silenced plants, demonstrating the limited spread of TuMV. Indeed, TuMV in new leaves from Nb*eIF4A*-silenced plants accumulated to much lower levels relative to TRV:00 control plants (S6B Fig). We also assessed the extent of NIb degradation in *NbeIF4A*-silenced plants. We transiently expressed NIb tagged with GFP in Nb*eIF4A*-silenced or control leaves by Agrobacterium-mediated infiltration. At 3 dpi, the accumulation of NIb-GFP was substantially reduced in Nb*eIF4A*-silenced leaves relative to TRV:00 control plants (S6C Fig). These results further support the hypothesis that silencing of Nb*eIF4A* activates autophagy.

## Inhibition of RSV infection in *NbeIF4A*-silenced plants requires activated autophagy

We recently reported that autophagy counters RSV infection by degrading the RSV p3 protein. Since silencing of Nb*eIF4A* activated autophagy, we reasoned that the autophagic degradation

of p3 should be enhanced and the activity of p3 as suppressor of RNA silencing should be hence weaken in Nb*eIF4A*-silenced cells, which would explain the limited spreading of RSV in these plants. To test our hypothesis, we expressed Nb4A-hairpin and GUS-hairpin individually in a single leaf of *N. benthamiana* plants (16c) by Agrobacterium-mediated infiltration. One day later, p3 and GFP were then expressed in the same leaf. At 3 dpi with p3 and GFP expression, green fluorescence in zones expressing 4A-hairpin was weak compared to that in zones expressing GUS-hairpin control (Fig 3A). We measured p3 and GFP accumulation by immunoblot analysis with anti-p3 and anti-GFP antibody, respectively. Both p3 and GFP accumulated to lower levels in zones expressing 4A-hairpin than in zones expressing GUS-hairpin (Fig 3B). In the control experiment, when GFP only was expressed in zones expressing 4A-hairpin or GUS-hairpin, the accumulation of GFP was not affected by 4A-hairpin (Fig 3C). These results demonstrate that the autophagic degradation of p3 is enhanced and the suppressor activity of RNA silencing is impaired in Nb*eIF4A*-silenced plants. These findings support the notion that the inhibition of RSV infection in Nb*eIF4A*-silenced plants is due to activated autophagy.

To further validate our hypothesis, we silenced the key autophagy gene Nb*ATG3* or Nb*ATG5*, together with Nb*eIF4A*, and monitored RSV infection in these plants. Nb*ATG3*- or Nb*ATG5*-silenced *N. benthamiana* plants showed no obvious change in phenotype but developed severe symptoms upon RSV infection (S7A–S7D Fig), and the virus spread more rapidly via systemic infection compared to control plants (S7E Fig). In agreement with the severity of infection, RSV CP accumulated to higher levels in Nb*ATG3* or Nb*ATG5*-silenced plants at 20 dpi compared to TRV:00 control plants (S7F Fig). These results are consistent with previous observations linking the degradation of viral proteins by autophagy with the degree of viral infection [16]. When Nb*eIF4A* was co-silenced with either Nb*ATG3* or Nb*ATG5*, typical symptoms of RSV appeared in all plants at 20 dpi, whereas only approximately 20% of Nb*eIF4A*-silenced plants showed mild RSV symptoms (Figs 3D, 3E and S8). Based on RSV CP accumulation, we determined that RSV infection was more pronounced in co-silenced plants than in Nb*eIF4A*-silenced plants, underscoring the compromised resistance to viral infection exhibited by the co-silenced plants (Figs 3F and S8). These results demonstrate that co-silencing of genes encoding key autophagy components compromises the inhibition of RSV infection in Nb*eIF4A*-silenced plants, further confirming the notion that the inhibition of RSV infection in Nb*eIF4A*-silenced plants depends on induced autophagic responses. However, we also noticed that silencing of Nb*ATG3* was not sufficient to cancel the effect of Nb*eIF4A* silencing on RSV accumulation, suggesting that Nb*eIF4A* might have another function during RSV infection.

## *Nb*eIF4A interacts with *Nb*ATG5

Next, we explored the mechanism by which *Nb*eIF4A intersects with other components of the autophagy pathway. To this end, we tested the potential interactions between *Nb*eIF4A and ATG3, ATG5, ATG6, ATG7, and ATG8, which were previously shown to function in the interplay between viruses and autophagy by yeast-two hybrid (Y2H) and bimolecular fluorescence complementation (BiFC) assays. Only *Nb*ATG5 interacted with *Nb*eIF4A (Figs 4A, 4B, S9A and S9B). We confirmed this interaction using coimmunoprecipitation (Co-IP) and firefly luciferase complementation imaging (LCI) assays (Fig 4C and 4D).

To map the domain(s) in *Nb*eIF4A responsible for its interaction with *Nb*ATG5, we generated a series of truncated *Nb*eIF4A mutants in accordance with the conserved domains in the protein: 4A(Δ40–68), 4A(71–241), 4A(252–413), 4A(71–187), 4A(195–241), and 4A(Δ187–195) (Fig 4E). In a BiFC assay, *Nb*ATG5 interacted with truncated *Nb*eIF4A proteins 4A(Δ40–

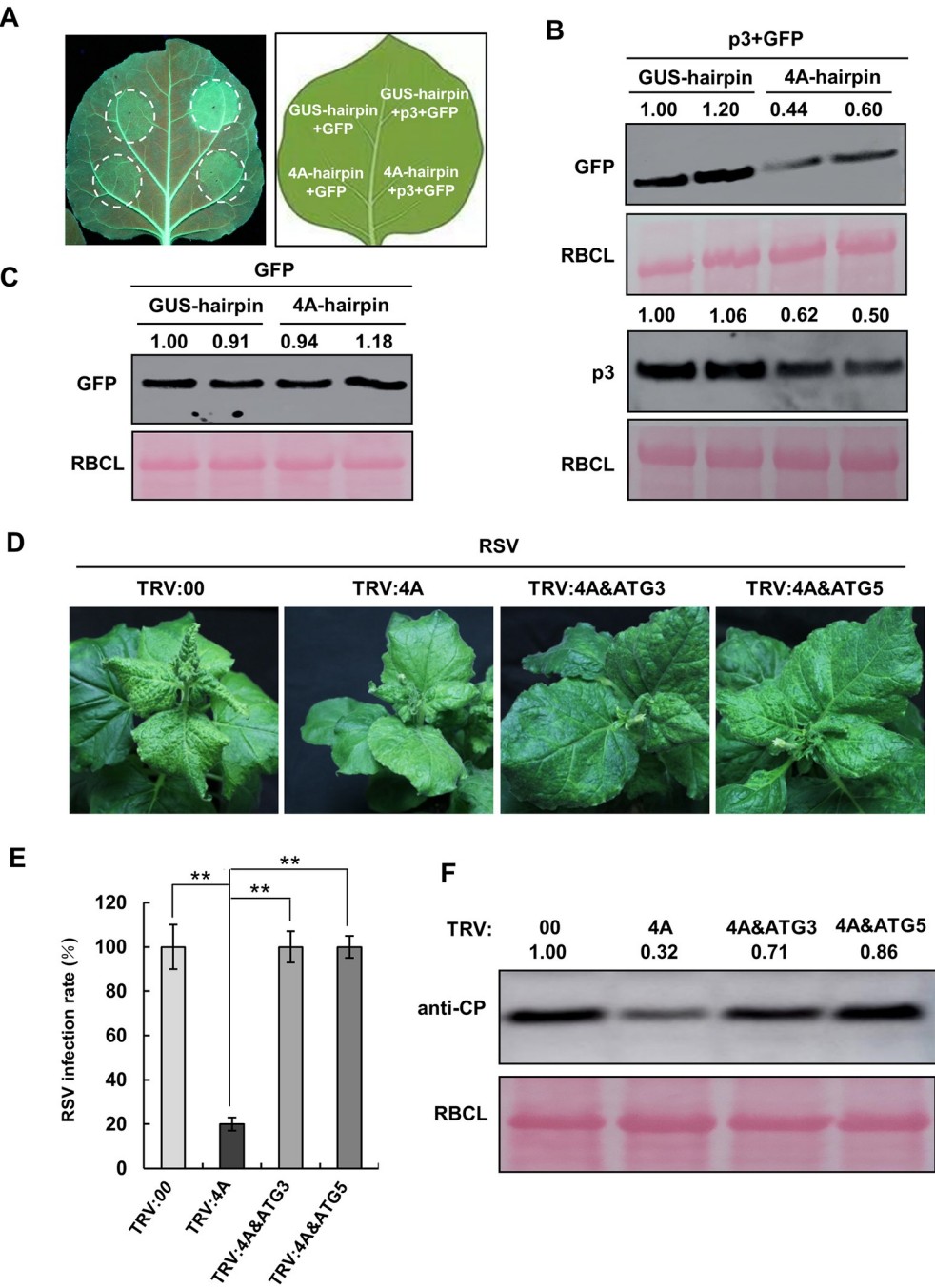

**Fig 3. Inhibition of RSV Infection in Nb*eIF4A*-Silenced Plants Requires Activated Autophagy. (A)** RNAi constructs expressing hairpin RNA of Nb*eIF4A* (4A-hairpin) and expressing hairpin RNA of β-glucuronidase (GUS-hairpin) were transiently expressed individually in one leaf of 16c plants for 1 day. p3 and (or) GFP were then expressed for additional 3 days. Green fluorescence was detected under UV. **(B)** Immunoblot analysis of protein expression probed with anti-p3 or anti-GFP antibody. Rubisco large subunit was used as a loading control. **(C)** Immunoblot analysis of GFP in a control experiment probed with anti-GFP antibody. **(D)** Development of RSV symptoms in co-silenced plants (TRV:4A&ATG3 and TRV:4A&ATG5), control plants (TRV:00), and Nb*eIF4A*-silenced plants (TRV:4A). **(E)** RSV infection rates in different plants. Error bars indicate standard error from three individual experiments. Twenty plants were used for each experiment. **(F)** Immunoblot analysis of RSV CP accumulation levels in the infected plants with anti-CP antibody. Band intensity was analyzed by ImageJ.

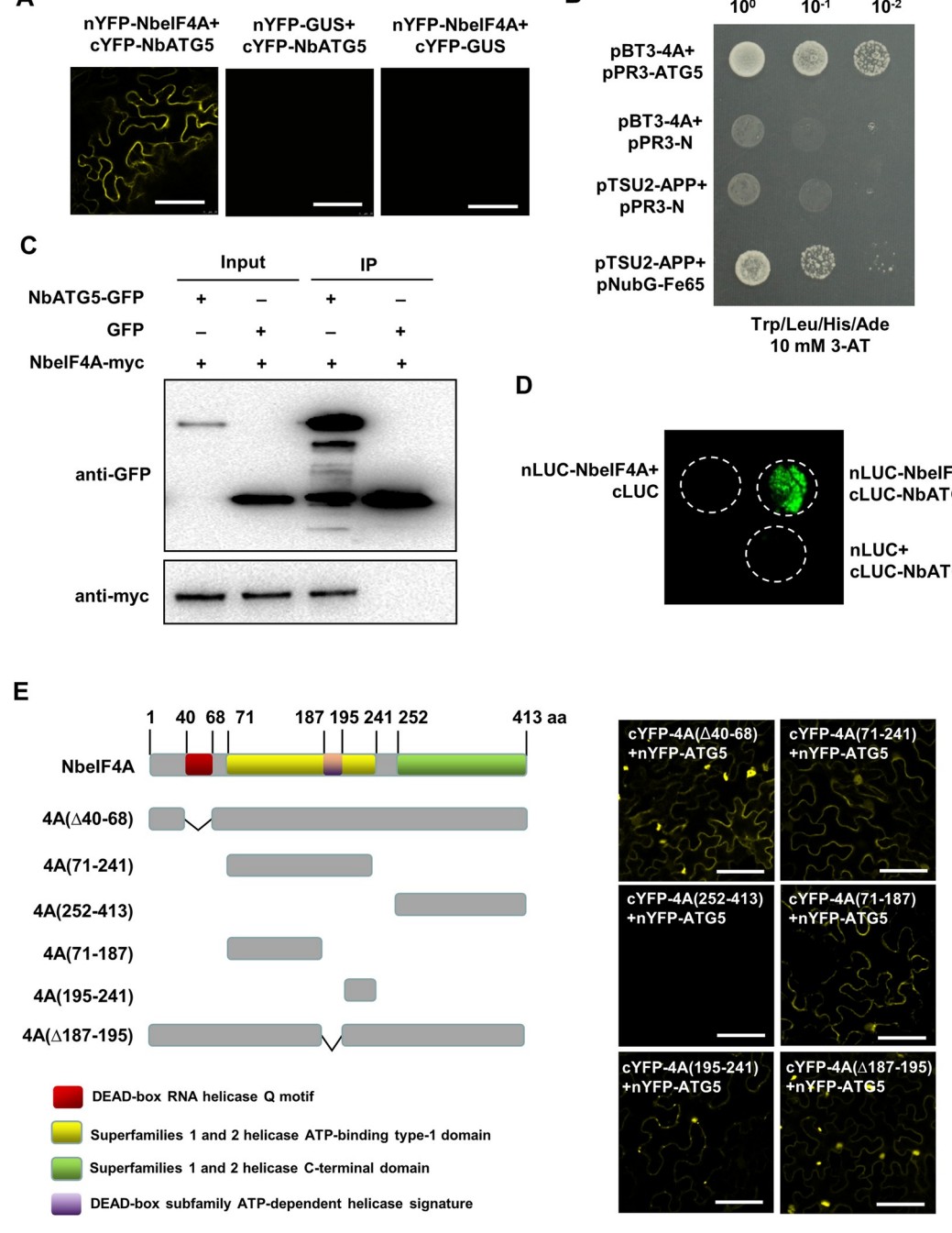

**Fig 4. *Nb*eIF4A interacts with *Nb*ATG5.** **(A)** *Nb*ATG5 and *Nb*eIF4A interaction, as tested by BiFC assay in cells expressing nYFP-4A (*Nb*eIF4A) and cYFP-ATG5 (*Nb*ATG5), and in control cells expressing nYFP-GUS and cYFP-ATG5, or cYFP-ATG5 and cYFP-GUS, under confocal microscopy. Bars, 20 μm. **(B)** *Nb*eIF4A interacts with NbATG5 in a Y2H assay. pBT3-4A (*Nb*eIF4A), pPR3-ATG5 (*Nb*ATG5), pPR3-N, positive control, or negative control constructs were co-transformed into yeast cells. The samples were plated onto selective medium lacking tryptophan, leucine, histidine, and adenine (SD/-Trp/-Leu/-His/-Ade) with 10 mM 3-AT. **(C)** Co-IP analysis of the interaction between *Nb*ATG5 and *Nb*eIF4A. Leaf tissues infiltrated with Agrobacterium cell suspensions carrying various constructs were harvested at 60 hpi. Total proteins were immunoprecipitated with anti-GFP beads. Input and immunoprecipitated protein (IP) were analyzed by immunoblot analysis with anti-GFP and anti-myc tag antibodies. **(D)** LCI assays of the interaction between *Nb*ATG5 and *Nb*eIF4A in *N. benthamiana*. Luminescence was measured in the infiltrated zone expressing nLUC-4A and cLUC-*Nb*ATG5, nLUC-4A and cLUC, cLUC-*Nb*ATG5, or nLUC. **(E)** BiFC assay of the potential interactions between *Nb*ATG5 and *Nb*eIF4A mutants. Schematic diagrams of the *Nb*eIF4A mutants are shown on the left. Bars, 20 μm.

68), 4A(71–187), 4A(71–241), 4A(Δ187–195), 4A(195–241), but not with 4A(252–413), suggesting that the ATP-binding type-1 domain from superfamily 1 and 2 helicases (amino acids 71–241 of *Nbe*IF4A) mediates its interaction with *Nb*ATG5 (Fig 4F).

## Overexpression of *Nbe*IF4A suppresses *Nb*ATG5-activated autophagy

Based on the finding that Nb*IF4A* silencing activated autophagy and that *Nbe*IF4A interacted with *Nb*ATG5, we hypothesized that *Nbe*IF4A likely suppresses the function of *Nb*ATG5, thereby inhibiting autophagy. We therefore examined the effects of *Nbe*IF4A overexpression on ATG5 function during autophagy. We tagged NbATG5 with red fluorescent protein (RFP) and co-expressed *Nb*ATG5-RFP with CFP-*Nb*ATG8f in *N. benthamiana* leaves by Agrobacterium-mediated infiltration. At 2.5 dpi, the number of autophagic bodies increased significantly in cells co-expressing both proteins compared to cells only expressing the control protein RFP, which is consistent with the activation of autophagy (Figs 5A, 5B and S10).

Next, we individually co-expressed myc-tagged *Nbe*IF4A; its mutant 4A(71–241), which retained the ability to interact with *Nb*ATG5; or its mutant 4A(252–413), which could no longer interact with *Nb*ATG5, with *Nb*ATG5-RFP and CFP-*Nb*ATG8f. Cells co-expressing myc-tagged *Nbe*IF4A or 4A(71–241) and *Nb*ATG5-RFP produced fewer autophagic bodies than cells expressing *Nb*ATG5-RFP only. However, the number of autophagic bodies in cells co-expressing myc-tagged 4A(252–413) and *Nb*ATG5-RFP was similar to that in cells expressing *Nb*ATG5-RFP alone (Figs 5A, 5B and S10). MDC staining produced similar results (Fig 5C and 5D). These results demonstrate that the accumulation of *Nbe*IF4A suppresses the ability of *Nb*ATG5 to activate autophagy. Moreover, such suppression is dependent on the interaction of *Nbe*IF4A with *Nb*ATG5.

We also examined the effects of *Nbe*IF4A on autophagy induced by methyl viologen (MV), which results in severe oxidative stress and induces autophagy [23,24]. Cells treated with MV showed a sharp increase in the number of autophagic bodies relative to untreated cells (S11 Fig). By contrast, MV treatment failed to induce the formation of autophagic bodies in cells expressing *Nbe*IF4A (S11 Fig), suggesting that *Nbe*IF4A also suppresses MV-activated autophagy.

During autophagy, ATG5 interacts with ATG12 to form ATG12–ATG5 conjugates. We wondered whether *Nbe*IF4A would interfere with such an interaction and hence suppress autophagy. To test this possibility, we examined the effects of expressing *Nbe*IF4A on the interaction between ATG5 and ATG12 via a BiFC assay. *Nb*ATG5 interacted with *Nb*ATG12 in the BiFC assay (S12A Fig). Furthermore, when *Nbe*IF4A-myc was coexpressed with these proteins in a BiFC assay, fluorescent signals significantly decreased, pointing to the impaired interaction between *Nb*ATG5 and *Nb*ATG12 (S12A Fig). Consistently, in a Co-IP assay, when *Nbe*IF4A was expressed, *Nb*ATG5 was immunoprecipitated, but it was difficult to immunoprecipitate *Nb*ATG12 in this assay (S12B Fig), indicating that expressing *Nbe*IF4A interferes with the interaction between *Nb*ATG5 and *Nb*ATG12.

Taken together, these results demonstrate that overexpressing *Nb*ATG5 activates autophagy and that expressing *Nbe*IF4A prevents this activation by competing with *Nb*ATG12 for interaction with *Nb*ATG5.

## Expression of vsiRNA-4A, which targets Nb*IF4A* mRNA for cleavage, induces autophagy

We previously showed that a siRNA derived from RSV RNA4 (vsiRNA-4A) targets Nb*IF4A* mRNA for cleavage [4]. Since silencing of Nb*IF4A* induced autophagy, we reasoned that expressing vsiRNA-4A should also induce autophagy by downregulating Nb*IF4A* expression.

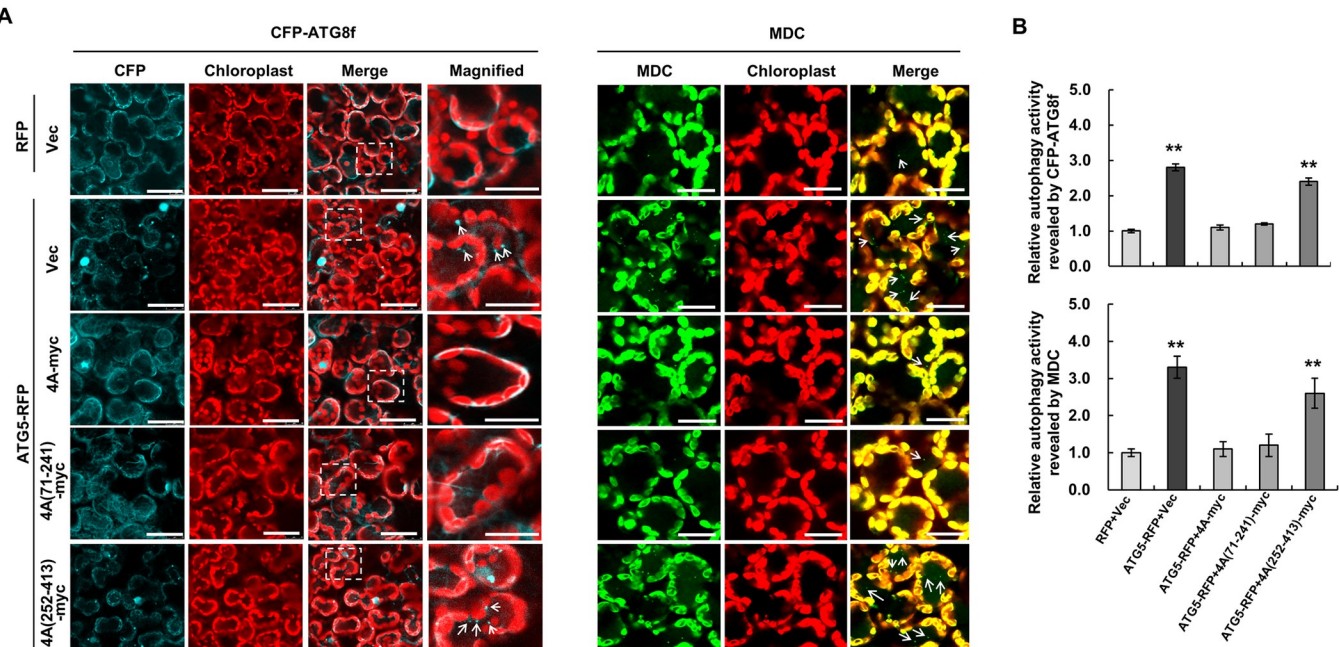

**Fig 5. Expression of NbeIF4A suppresses NbATG5-activated autophagy.** (A) Representative confocal images of autophagic activity revealed by the autophagy marker CFP-*Nb*ATG8f or MDC staining. RFP or ATG5-RFP was co-expressed with 4A-myc, 4A(71–241)-myc, 4A(252–413)-myc, and empty vector (Vec) in *N. benthamiana* leaves by Agrobacterium-mediated infiltration. Cyan: CFP-ATG8f signal; green: MDC-stained structures; red: chlorophyll autofluorescence. Bars, 20 μm. Magnified area is indicated by a white dashed box. (B) Relative autophagic activity, normalized to that of cells expressing RFP and empty vector, which was set to 1.0. Quantification of CFP-ATG8f-labeled or MDC-stained autophagic foci per cell was performed by counting autophagic bodies to calculate the autophagic activity. Over 150 cells per treatment were used for quantification. Error bars indicate standard error from three individual experiments. Asterisks indicate significant differences by Student's *t*-test compared to control cells expressing RFP and Vec (**, $p < 0.01$). (C) Immunoblot analysis of the accumulation of proteins examined in this experiment.

To test this hypothesis, we expressed vsiRNA-4A via the same method used to express artificial miRNAs in *N. benthamiana* leaves [4]. At 2.5 dpi, cells co-expressing vsiRNA-4A and CFP-*Nb*ATG8f showed a 40% decrease in Nb*eIF4A* transcript levels relative to control cells co-expressing a control small RNA and CFP-*Nb*ATG8f, demonstrating the silencing of Nb*eIF4A* by vsiRNA-4A (S13 Fig). Using confocal microscopy, we observed autophagosomes and autophagic bodies in cells co-expressing vsiRNA-4A and CFP-*Nb*ATG8f, but fewer of these structures in control cells (Fig 6A and 6B). We independently validated these results by MDC staining (Fig 6C and 6D). These results demonstrate that expressing vsiRNA-4A induces autophagy.

To further support our hypothesis that the induction of autophagy depends on the downregulation of Nb*eIF4A*, we co-expressed vsiRNA-4A and CFP-*Nb*ATG8f in the leaves of transgenic *N. benthamiana* overexpressing Nb*eIF4A* generated in our previous study [4]. The expression of vsiRNA-4A failed to reduce Nb*eIF4A* transcript levels in transgenic plants to lower levels than observed in the wild type and did not induce autophagy (Fig 6E and 6F). We obtained similar results using MDC staining (Fig 6G and 6H). These results demonstrate that the expression of vsiRNA-4A induces autophagy via the downregulation of Nb*eIF4A*.

## Rice eIF4A interacts with *Os*ATG5 and suppresses *Os*ATG5-activated autophagy

Finally, we looked for the rice (*Oryza sativa*) ortholog of Nb*eIF4A*. Os*eIF4A* (Os06g0701100) shows 81% nucleotide sequence identity with Nb*eIF4A*, and the encoded protein shares 74.6%

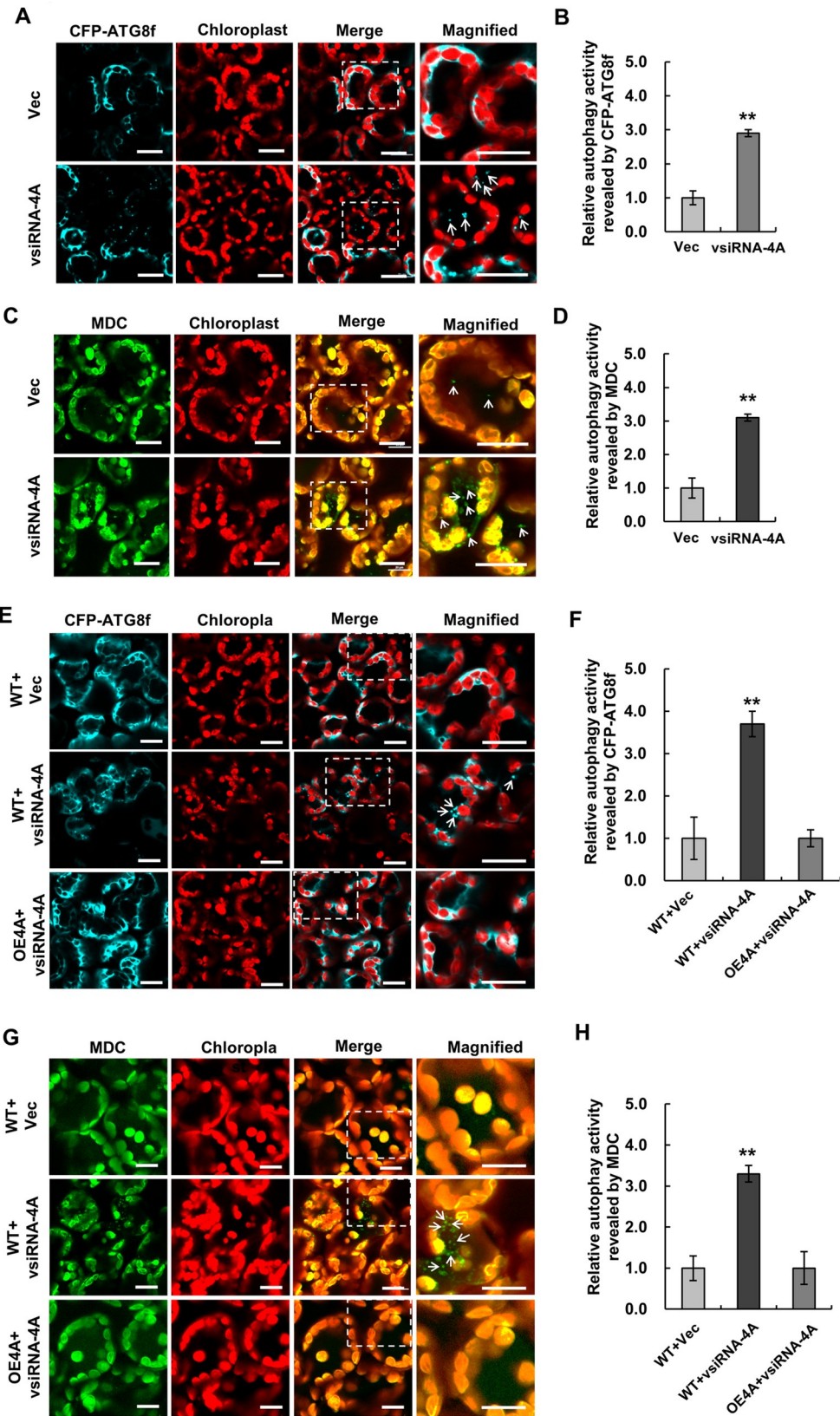

**Fig 6. Expression of vsiRNA-4A, which targets *NbeIF4A* mRNA for cleavage, induces autophagy. (A–D)** Autophagy induced by vsiRNA-4A expression. Representative confocal images of autophagic activity revealed by the autophagy marker CFP-*Nb*ATG8f **(A)** or MDC staining **(C)**. Relative autophagic activity in treated cells was normalized to that of control cells expressing empty vector (Vec), which was set to 1.0. Quantification of CFP-*Nb*ATG8f-labeled **(B)** or MDC-stained **(D)** autophagic foci per cell was performed by counting autophagic bodies to calculate the autophagic activity. Over 150 cells per treatment were used for quantification. Bars indicate standard error from three individual experiments. Asterisks indicate significant differences by Student's *t*-test compared to the control (**, $p<0.01$). **(E–H)** Expression of vsiRNA-4A fails to induce autophagy in transgenic plants overexpressing Nb*IF4A* (OE4A). Representative confocal images of autophagic activity revealed by the autophagy marker CFP-*Nb*ATG8f **(E)** or MDC staining **(G)**. Relative autophagic activity in treated cells was normalized to that in control wild-type cells expressing empty vector (Vec), which was set to 1.0. Quantification of CFP-*Nb*ATG8f-labeled **(F)** or MDC-stained **(H)** autophagic foci per cell was performed by counting autophagic bodies to calculate autophagic activity. More than 150 cells per treatment were used for quantification. Bars indicate standard error from three individual experiments. Asterisks indicate significant differences by Student's *t*-test compared to control cells expressing empty vector (**, $p<0.01$).

amino acid sequence identity with *Nb*eIF4A (S14A and S14B Fig). In addition, *Os*eIF4A mRNA is also targeted by vsiRNA-4A for cleavage (S15 Fig). We therefore tested whether *Os*eIF4A performs similar functions to *Nb*eIF4A. Indeed, *Os*eIF4A interacted with rice ATG5 (*Os*ATG5) (Fig 7A and 7B). *Os*ATG5 shares higher sequence identity with *Nb*ATG5 in the N-terminus of the protein (S16 Fig). Transient expression of *Os*ATG5 in *N. benthamiana* activated autophagy (Fig 7C–7E). Moreover, co-expression of *Os*eIF4A effectively suppressed autophagy activated by *Os*ATG5 or *Nb*ATG5 (Figs 7C–7E and S17). These results suggest that *Os*eIF4A plays a similar role to *Nb*eIF4A, i.e., its transcript is targeted by vsiRNA-4A and suppresses autophagy by interacting with *Os*ATG5. These findings highlight the functional conservation of eIF4A in the regulation of autophagy in *N. benthamiana* and rice.

## Discussion

We previously reported that the downregulation of Nb*eIF4A* transcript by vsiRNA-4A contributes to viral symptoms in RSV-infected *N. benthamiana*. Here, we uncovered the biological function of downregulated Nb*eIF4A*: it regulates the autophagic responses that normally inhibit RSV infection, pointing to the complex roles of *Nb*eIF4A during RSV infection. eIF2α, a member of the eukaryotic translation initiation factor (eIF) family, is associated with endoplasmic reticulum (ER) stress-induced autophagy in mammalian cells [25,26]. eIF2α can be phosphorylated by the PKR-like ER kinase PERK [25]. PERK/eIF2α phosphorylation is critical for the conversion of microtubule-associated protein 1 (MAP1) light chain 3 (LC3) from LC3-I to LC3-II, which plays a key role in autophagy [26]. eIF5A was recently identified as a key factor required for the lipidation of members of the ATG8 family of proteins as well as autophagosome formation via translation of the E2 ubiquitin ligase-like ATG3 protein in mammalian cells [27,28]. Yet, little is known about the roles of eIFs is plant autophagy. Our results provide the first evidence that eIF4A functions in regulating antiviral autophagy in plants.

eIF4A is thought to use the energy from ATP hydrolysis to unwind mRNA structures and, together with other components, prepare mRNA templates for ribosome recruitment during translation initiation [1]. The current results suggest that eIF4A functions as a negative regulator of autophagy that plays a role in protein metabolism by degrading proteins. These findings point to the intricate roles of eIF4A in protein production: not only does it function in translation initiation, but also in inhibiting autophagy, a mechanism for protein degradation. We also noticed that although silencing of Nb*eIF4A* caused a dwarf phenotype in *N. benthamiana*, which is consistent with findings for *B. distachyon* [3], it did not significantly affect protein

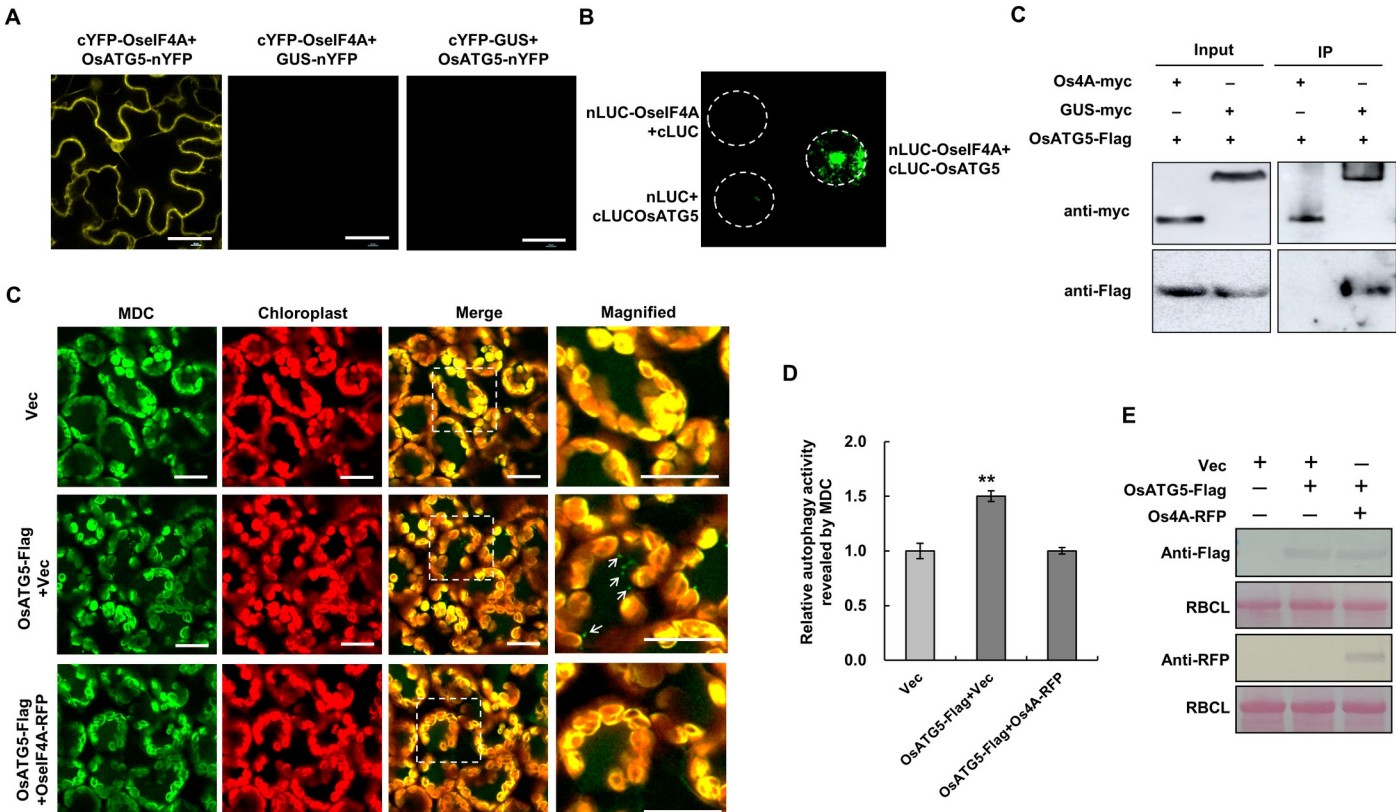

**Fig 7. Rice eIF4A (OseIF4A) Interacts with *Os*ATG5 and Suppresses *Os*ATG5-Activated Autophagy.** (A–C) Interaction of *Os*eIF4A with *Os*ATG5, as assessed by BiFC (A), LIC (B), and Co-IP (C) assays. (D, E) Representative confocal images of autophagic activity in *N. benthamiana* cells expressing *Os*ATG5 and co-expressing *Os*ATG5 and *Os*eIF4A revealed by MDC staining (D), and relative normalized autophagic activity (E). Green: MDC-stained structures; red: chlorophyll autofluorescence. Relative autophagic activity was normalized to that of cells expressing empty vector. Quantification of MDC-stained structures (autophagic foci per cell) was performed by counting autophagic bodies to calculate autophagic activity. More than 150 cells per treatment were used for quantification. Error bars indicate standard error from three individual experiments. Asterisks indicate significant differences by Student's *t*-test (**, $p < 0.01$) compared to the control. (F) Immunoblot analysis of the accumulation of proteins in Agrobacterium-infiltrated plants.

translation (S7 Fig). We suggest that other components might complement the function of silenced Nb*eIF4A* in translation.

Several studies have shown that autophagy is regulated during viral infection. Xu et al. demonstrated that plant Bax inhibitor-1 (BI-1) interacts with ATG6 to regulate autophagy [29]. Silencing of BI-1 reduced the autophagic activity induced by N gene-mediated resistance to TMV, and overexpressing plant BI-1 increased autophagic activity [29]. The cytosolic glyceraldehyde-3-phosphate dehydrogenase (GAPC) negatively regulates autophagy by interacting with ATG3 [20]. Silencing of *GAPCs* significantly activated ATG3-dependent autophagy, while overexpressing *GAPCs* suppressed autophagy in *N. benthamiana* plants [20]. In the current study, we determined that Nb*eIF4A* suppresses ATG5 activity by interfering with the ATG5–ATG12 interaction, thus demonstrating its role as a new negative regulator of autophagy in plants. In addition, the current and previous results suggest that autophagy factors may be targeted by other host factors to regulate autophagy [16,20,29]. It is expected that more host factors will be identified in the future, which will help further dissect the mechanisms regulating autophagy.

Autophagy plays essential defense roles against CaMV, TuMV, and CLCuMuV infection [12–15,18]. Moreover, accumulating evidence suggests that autophagy benefits the virus by contributing to infection [7,30]. Indeed VSR P0 from *Polerovirus* triggers the autophagic

degradation of ARGONAUTE 1 (AGO1), a key component of the antiviral RNA silencing pathway [31]. VPg of TuMV mediates the degradation of SUPPRESSOR OF GENE SILENCING 3 (SGS3) via ubiquitination and autophagy to promote virus infection [32]. βC1 of *Tomato yellow leaf curl China virus* (TYLCCNV) regulates *CaM* expression, thereby mediating the autophagic degradation of *Nb*SGS3 in *N. benthamiana* [33]. Virus-induced vesicles enriched with ATG8f provide an alternative site for *Bamboo mosaic virus* (BaMV) RNA replication or a shelter from the host silencing mechanisms [34]. Together, these findings reveal that autophagy can be manipulated or even exploited by viruses for infection, highlighting the complex roles of autophagy in plant–virus interactions [9,10].

Our previous and current results demonstrate that autophagy also functions in defense against RSV infection, illustrating the complex role of autophagy during plant–RSV interactions [16]. Fu et al. reported that the RSV-encoded NSvc4 protein interferes with the S-acylation of remorin, which normally prevents viral movement, thus inducing its degradation through the autophagy pathway [35]. This observation also demonstrates that autophagy can function indirectly during plant–RSV interactions to benefit RSV infections. This might also represent a common way in which autophagy participates in plant–virus interactions. For example, AGO1 and SGS3 are degraded via autophagy during *Polerovirus*, TuMV, or TYLCCNV infection [31–33]. This observation could also help explain why RSV retained a sequence that can induce autophagy. Perhaps the induced autophagy benefits RSV by facilitating its movement.

Infection with CLCuMuB, TuMV, and RSV induces autophagy, although the underlying mechanism is not well understood [14,15]. Upregulated expression of *ATGs* it thought to be associated with the induction of autophagy in TuMV- and RSV-infected plants [12]. Ismayil et al. reported that CLCuMuB βC1 interacts with the negative autophagic regulator GAPC to induce autophagy in plants, suggesting a mechanism by which CLCuMuB induces autophagy [19,20]. Here we demonstrated that *Nbe*IF4A negatively regulates antiviral-associated autophagy in *N. benthamiana*. Moreover, a vsiRNA that target s*Nbe*IF4A mRNA for cleavage induces autophagy. These findings suggest a mechanism in which a negative regulator of antiviral autophagy (*Nbe*IF4A) sacrifices itself to induce autophagy against viral infection by allowing its transcripts to be recognized and cleaved by vsiRNAs.

Mounting evidence indicates that vsiRNAs play roles in the interaction between viruses and their host plants by regulating host gene expression [36–40]. For example, vsiRNA-20 from *Chinese wheat mosaic virus* (CWMV) affects pyrophosphate hydrolysis and H(+) concentrations in CWMV-infected wheat (*Triticum aestivum*) cells by regulating the mRNA accumulation of vacuolar $H^+$-PPases to create a more favorable cellular environment for CWMV replication [41]. The current results support the indirect participation of vsiRNAs in the regulation of autophagy.

Based on these observations, we propose a model for the possible role of autophagy in RSV infection. According to this model, eIF4A functions as a negative regulator to suppress autophagy in plants by competitively interacting with ATG5 to interfere with its interaction with ATG12. Upon RSV infection, *eIF4A* mRNA becomes targeted by vsiRNAs for cleavage. The resulting downregulation of eIF4A releases ATG5 to activate autophagy, which in turn prevents RSV infection by degrading the p3 protein, a suppressor of antiviral RNA silencing (Fig 8). Perhaps this mechanism could be exploited to design ways to improve plant virus resistance, for example, by editing *eIF4A*.

## Materials and methods

### Plant growth conditions and viral inoculation

We grew *Nicotiana benthamiana* plants in pots in a growth room at 24˚C and 60% relative humidity under a 16h-light/8h-dark cycle. pTRV1 and pTRV2 or its derivatives were

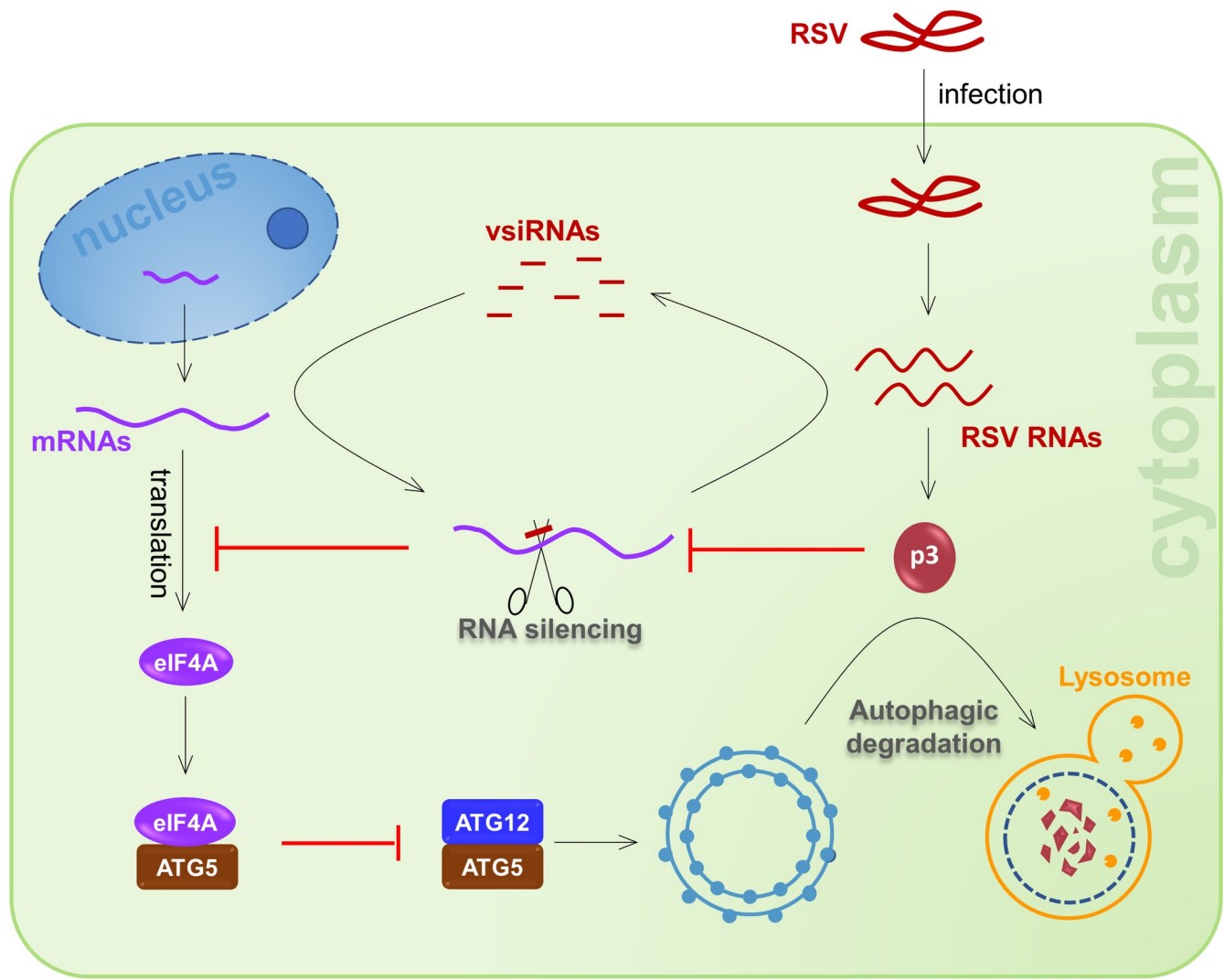

**Fig 8. Proposed model for the induction of autophagy and its antiviral roles in RSV infection.** In plants, eIF4A functions as a negative regulator that suppresses autophagy by interacting with ATG5. Upon RSV infection, *eIF4A* mRNA is targeted by vsiRNAs for cleavage. The resulting downregulation of eIF4A releases ATG5 to activate autophagy, which inhibits RSV infection by degrading p3 protein, a suppressor of antiviral RNA silencing.

introduced into Agrobacterium (*Agrobacterium tumefaciens*) strain GV3101. For the VIGS assays, we resuspended Agrobacterium cell pellets bearing pTRV1 and pTRV2-VIGS (TRV:00, TRV: Nb*eIF4A*, TRV: Nb*ATG3*,TRV: Nb*ATG5*, TRV: Nb*eIF4A*+*NbATG3*, TRV: Nb*eIF4A* +*NbATG5*) in infiltration buffer (10 mM MgCl₂, 10 mM MES (pH 5.6), and 100 μM acetosyringone) and mixed the bacterial suspensions at a 1:1 ratio. Following 2–4 h incubation at room temperature, the mixed Agrobacterium cell suspensions were infiltrated into *N. benthamiana* leaves from plants at the 5–6-leaf stage. Silenced phenotypes appeared in the upper leaves 10–15 d post infiltration (dpi).

For RSV inoculation, we mechanically inoculated the middle of each *N. benthamiana* leaf with sap from RSV-infected rice. Symptoms were photographed with a Canon 550D digital camera at the indicated times. For TuMV-GFP inoculation, we infiltrated an Agrobacterium cell suspension carrying the TMV-GFP plasmid into *N. benthamiana* leaves as previously described [25,26]. GFP was imaged under UV light.

## Plasmid constructs

We PCR-amplified mutant Nb*eIF4A* and Nb*eIF4A* sequences using the cloned cDNA as template and cloned the resulting PCR products into the pJG045 vector. We generated DNA fragments for Nb*eIF4A-GFP*, Nb*eIF4A-myc*, *nLUC*-Nb*eIF4A*, *nYFP*-Nb*eIF4A*, *cYFP*-Nb*eIF4A* mutants, and Nb*eIF4A* mutants-myc by overlapping PCR. We PCR-amplified the Nb*ATG5* cDNA and cloned the PCR product into the pJG045 vector. We obtained DNA fragments for Nb*ATG5-GFP*, Nb*ATG5-RFP*, *nYFP*-Nb*ATG5*, *cYFP*-Nb*ATG5*, and *cLUC*-Nb*ATG5* by overlapping PCR. We amplified the Os*eIF4A* andOs*ATG5* cDNAs by PCR and cloned the PCR product into the pJG045 vector. We generated the DNA fragments for *cYFP*-Os*eIF4A*, Os*eIF4A-myc*, Os*eIF4A-RFP*, *nLUC*-Os*eIF4A*, Os*ATG5-nYFP*, Os*ATG5-Flag*, and *cLUC*-Os*ATG5* by overlapping PCR.

To construct a TRV-based recombinant VIGS vector containing Nb*eIF4A*, Nb*ATG3*, or Nb*ATG5*, we PCR-amplified a partial fragment for each gene using the appropriate primer pair and cloned the PCR product into the pTRV2-lic vector. Two fusion DNA fragments for Nb*eIF4A*+*NbATG3* and Nb*eIF4A*+*NbATG5* were obtained by overlapping PCR and cloned into the pTRV2-lic vector.

The primers used for these vectors are listed in S1 Table.

## Bi-molecular Fluorescence Complementation (BiFC) assays

For the BiFC experiments, we infiltrated fully expanded leaves of three-week-old *N. benthamiana* plants with Agrobacterium strain (GV3101) containing constructs at a cell density $OD_{600}$ = 0.5. At 36–72 h after Agrobacterium-mediated infiltration, we examined the epidermal cells of 1-cm infiltrated leaf explants by confocal microscopy (Leica TCS SP5, Mannheim, Germany).

## Co-immunoprecipitation (Co-IP) and immunoblot analysis

We harvested infiltrated leaves at 48–72 h after transient Agrobacterium-mediated infiltration and extracted total proteins from frozen, ground tissue by mixing with 3 Laemmli buffer at a 1:1 ratio (w/v) for 5 min. Cell debris was collected by centrifugation at 12,000 $g$ for 5 min at 4˚C, after which we determined the protein concentration in the supernatant by Bio-Rad Protein assay. Equal amounts of total protein were aliquoted, brought to 40 μL with protein extraction buffer, and mixed with 10 μL 5× SDS-PAGE loading buffer. After boiling the samples for 10 min at 100˚C, we quickly chilled the total protein lysate on ice for 5 min, centrifuged the sample at 12,000 $g$ for 1 min at 4˚C, and separated the proteins on SDS-PAGE gels. We performed immunoblotting with anti-myc (Sigma, St. Louis, USA), anti-RFP (Sigma, St. Louis, USA), and anti-GFP (Sigma, St. Louis, USA) primary antibodies and anti-rabbit or anti-mouse (Sigma, St. Louis, USA) secondary antibody at 1:10,000 dilution. Antibodies of TuMV CP, RSV CP and P3 proteins were prepared in our laboratory.

For IP, protein extracts were incubated with anti-GFP antibody for 4 h at 4˚C. The beads were washed six times with ice-cold IP buffer at 4˚C. The IP samples were analyzed by SDS-PAGE, immunoblotted using anti-myc antibody, and detected using Pierce ECL Western Blotting Substrate (Amersham Image 680).

## Luciferase Complementation Imaging (LCI) assays

For the LCI assays, we introduced the plasmid combinations into *N. benthamiana* leaves by Agrobacterium-mediated infiltration. We kept infiltrated leaves in the dark for 24 hpi and then treated them with white light for 48 hpi. We detached the leaves at 72 hpi and sprayed

them with 1 mM luciferin. After keeping the materials in the dark for 5 min to quench the chlorophyll auto-fluorescence, we collected the luciferase signal with a low-light cooled CCD imaging apparatus.

## RNA extraction and RT-qPCR

We extracted total RNA from *N. benthamiana* leaf tissues using TRIzol reagent and treated the RNA with RNase-free DNase I (TaKaRa, China) to remove potential DNA contamination. We performed first-strand cDNA synthesis using 1 μg of total RNA with oligo(dT)$_{12-18}$ primers using a Titanium One-Step RT-PCR Kit (TaKaRa, Japan). Primers used for RT-qPCR are listed in S1 Table.

## Confocal Microscopy and Transmission Electron Microscopy (TEM)

We performed confocal microscopy using a Leica TCS SP5 (Mannheim, Germany). We introduced all combinations tested into *N. benthamiana* leaves by Agrobacterium-mediated infiltration. For GFP, RFP, CFP, and YFP imaging, the fluorescent proteins were excited using the LD laser line at 488 nm (GFP), 514 nm (RFP), 405 nm (CFP), and 514 nm (YFP). Detection bands were optimized for each fluorophore group to avoid emission bleeding.

For monodansylcadaverine (MDC) staining, we infiltrated leaves with 100 mM E-64d (Sigma) and incubated the leaves in the dark for 8–12 h. We excised the infiltrated parts of leaves and immediately vacuum-infiltrated them with 50 mM MDC (Sigma) for 8–14 min, followed by two washes with phosphate buffered saline (PBS). MDC was excited at a wavelength of 405 nm and detected from 450 to 550 nm. Chlorophyll autofluorescence was excited at 543 nm and detected from 580 to 700 nm.

To observe autophagosomes using the autophagy marker CFP-ATG8f, we introduced CFP-ATG8f into leaves of 5–6 leaf stage *N. benthamiana* plants by Agrobacterium-mediated infiltration. After 48 h, we infiltrated the leaves with 100 mM E-64d (Sigma) and incubated them in the dark for 8–12h. We observed the leaves by confocal microscopy with an excitation of 405 nm and the emission captured at 454 to 581 nm (Leica TCS SP5, Mannheim, Germany).

For electron microscopy, we cut leaves into small fragments (1–2 mm$^2$) and infiltrated them with 0.1 M PBS buffer containing 2.5% glutaraldehyde for fixation. We post-fixed the samples in 2% OsO$_4$, followed by dehydration in ethanol and acetone, before embedding in Spurr resin (SPI Supplies). We cut the sections with a diamond knife on an ultramicrotome (EM UC7; Leica, Germany) and collected them on copper grids. The sections were double-stained with uranyl acetate and lead citrate before examination.

## Accession numbers

GenBank accession numbers of the genes and viral sequences analyzed in this study are as follows: Nb*eIF4A* (XM019409915), Nb*ATG2* (KU561373), Nb*ATG3* (KX369396), Nb*ATG5* (KX369397), Nb*ATG6* (AY701316), Nb*ATG7* (KX369398), Nb*PI3K* (KX120977), Nb*ATG8f* (KU561372), TuMV (NC002509), Nb*Actin* (AY179605), Os*eIF4* (XM026026114), Os*ATG5* (XM015771963). The nucleotide sequence of Nb*ATG12* cloned here has 100% identity to that of *Nicotiana tabacum* ATG12 (XM016602097).

## Supporting information

**S1 Fig. RSV infection in TRV:00- and TRV:4A-infected *N. benthamiana*.** Typical RSV symptoms such as stunting and leaf-twisting with yellow mosaicism appeared on all TRV:00

control plants, while only 25% of RSV-inoculated Nb*eIF4A*-silenced plants showed yellow mosaicism on leaves at 20 dpi, which was less pronounced than the control. Silencing of Nb*eIF4A* caused leaf-twisting, which may interfere with the observation of RSV symptoms. The most obvious symptom of RSV infection is yellow and mosaic leaves, which was attenuated in Nb*eIF4A*-silenced plants.
(TIF)

**S2 Fig. Confirmation of RSV infection in TRV:00- and TRV:4A-infected *N. benthamiana*.**
**(A)** RT-PCR to analyze systemic RSV infection. Results from three replicates (R1, R2 and R3) are shown. Twenty plants per replicate were subjected to infection by TRV:4A or TRV:00. **(B)** Immunoblot analysis of RSV CP accumulation in RSV-infected Nb*eIF4A*-silenced plants in three replicates (R1, R2, and R3). RSV-infected TRV:00-infected plants were used as a control. Band intensity was analyzed by ImageJ.
(TIF)

**S3 Fig. Relative expression levels of autophagy-related genes in Nb*eIF4A*-silenced *N. benthamiana* plants.** RT-qPCR was used to measure the relative transcript level of genes in the autophagy pathway. Nb*Actin* served as the internal control. Error bars indicate standard error from three individual experiments. Asterisks indicate significant differences by Student's *t*-test compared to the control (*, $p<0.05$; **, $p<0.01$).
(TIF)

**S4 Fig. Silencing of Nb*eIF4A* does not affect protein translation. (A)** Immunoblot analysis of GFP expressed in Nb*eIF4A*-silenced cells (4A-hairpin), Nb*eIF6A*-silenced cells (6A-hairpin), or non-silenced cells (GUS-hairpin). Rubisco large subunit was used as a loading control. Silencing of Nb*eIF6A* was used as positive control, as it impaired protein translation. **(B)** RT-qPCR analysis of Nb*eIF6A* expression in Nb*eIF6A*-silenced cells. Asterisks indicate significant differences by Student's *t*-test (***, $p < 0.005$) compared to the control. **(C)** RT-qPCR analysis of Nb*eIF4A* mRNA levels in *NbeIF4A*-silenced cells. Asterisks indicate significant differences by Student's *t*-test (***, $p < 0.005$) compared to the control. **(D)** RT-qPCR analysis of *GFP* mRNA levels in Nb*eIF6A*-silenced cells. Asterisks indicate significant differences by Student's *t*-test (*, $p < 0.05$) compared to the control. **(E)** RT-qPCR analysis of *GFP* mRNA levels in Nb*eIF4A*-silenced cells. Asterisks indicate significant differences by Student's *t*-test (***, $p < 0.005$) compared to the control.
(TIF)

**S5 Fig. Relative expression levels of Nb*ATG3* and Nb*eIF4A* in co-silenced plants. (A)** Relative expression level of *NbATG3* in *NbATG3*- and *NbeIF4A* co-silenced *N. benthamiana* plants. **(B)** Relative expression level of Nb*eIF4A* in Nb*ATG3* and Nb*eIF4A* co-silenced *N. benthamiana* plants, as determined by RT-qPCR. *NbActin* served as the internal control. Error bars indicate standard error from three individual experiments. Asterisks indicate significant differences by Student's *t*-test compared to the control (*, $p<0.05$; **, $p<0.01$).
(TIF)

**S6 Fig. Silencing of Nb*eIF4A* inhibits *Turnip mosaic virus* infection of *N. benthamiana* plants and promotes the degradation of viral NIb protein. (A)** TuMV-GFP infection in Nb*eIF4A*-silenced (TRV:4A) plants. Non-silenced (TFV:00) plants were used as controls. In TuMV-GFP-inoculated leaves of Nb*eIF4A*-silenced plants, the number of infection foci decreased significantly at 3 dpi. Fluorescent foci under UV illumination indicate GFP-positive infection foci. At 6 dpi, intensive fluorescence was observed in the top leaves of control plants, indicating the systemic infection of plants by TuMV-GFP. Weak fluorescence appeared in the

top leaves of Nb*eIF4A*-silenced plants. **(B)** Immunoblot analysis of TuMV-GFP accumulation in TRV:00- and TRV:4A-infected *N. benthamiana* plants with antibody of TuMV CP. **(C)** Immunoblot analysis of the accumulation of GFP-fused TuMV NIb protein transiently expressed in Nb*eIF4A*-silenced and control leaves by Agrobacterium-mediated infiltration at 2.5 dpi. Band intensity was analyzed by ImageJ.
(TIF)

**S7 Fig. RSV infection in Nb*ATG3* and Nb*ATG5*-silenced *N. benthamiana*. (A)** Silencing of Nb*ATG3* and Nb*ATG5* individually did not cause obvious phenotypes in *N. benthamiana*. **(B, C)** Decreased expression of Nb*ATG3* and Nb*ATG5* in the corresponding silenced plants, as revealed by RT-qPCR, indicating the effective silencing of these genes. *NbActin* served as the internal control. Error bars indicate standard error from three individual experiments. Asterisks indicate significant differences by Student's *t*-test compared to the control (*, $p<0.05$; **, $p<0.01$). **(D)** RSV symptoms in Nb*ATG3*-or Nb*ATG5*-silenced plants at 20 dpi. **(E)** Systemic RSV infection in silenced plants and control plants. Error bars indicate standard error from three individual experiments. Twenty plants were used for each experiment. **(F)** Immunoblot analysis of RSV accumulation in silenced plants. Band intensity was analyzed by ImageJ.
(TIF)

**S8 Fig. Co-silencing of Nb*ATG3* or Nb*ATG5* with Nb*eIF4A* in *N. benthamiana* plants and immunoblot analysis of RSV CP accumulation in the silenced plants. (A)** Phenotypes of Nb*ATG2*, Nb*ATG3*, or Nb*ATG5* and Nb*eIF4A* co-silenced plants. **(B–D)** Decreased expression levels of Nb*ATG3* **(B)**, Nb*ATG5* **(C)** and Nb*eIF4A* **(D)** in the corresponding co-silenced plants, as revealed by RT-qPCR, indicating the efficient silencing of these genes. Nb*Actin* served as the internal control. Error bars indicate standard error from three individual experiments. Asterisks indicate significant differences by Student's *t*-test compared to the control (*, $p<0.05$; **, $p<0.01$). **(E)** Immunoblot analysis of RSV CP accumulation levels in the infected plants in the other two repeated experiments. Band intensity was analyzed by ImageJ.
(TIF)

**S9 Fig. Y2H and BiFC identify *Nb*ATG5 as an interactor of *Nb*eIF4A. (A)** *Nb*ATG5, but not other ATGs, interacted with *Nb*eIF4A. The samples were plated onto selective medium lacking tryptophan and leucine (SD/-Trp/-Leu) and tryptophan, leucine, histidine, and adenine (SD/-Trp/-Leu/-His/-Ade) with 10 mM 3-AT. **(B)** BiFC assay confirming the results from Y2H. Bars, 20 μm.
(TIF)

**S10 Fig. Immunoblot analysis confirming the expression of proteins used in the analysis of *Nb*eIF4A-mediated inhibition of ATG5-activated autophagy.** Expression of proteins used in the analysis of NbeIF4A-mediated inhibition of ATG5-activated autophagy was demonstrated using antibodies to the corresponding tags.
(TIF)

**S11 Fig. Expression of *Nb*eIF4A suppresses MV-induced autophagy. (A, C)** Representative confocal images of dynamic autophagic activity observed in MV-treated cells, empty vector control (EV), and myc-tagged *Nb*eIF4A-expressing cells, revealed by the specific autophagy marker CFP-*Nb*ATG8f **(A)** or MDC staining **(C)**. Cyan: CFP-*Nb*ATG8f; green: MDC-stained structures; red: chlorophyll autofluorescence. Bars, 10 μm. **(B, D)** Relative autophagic activity in treated cells, normalized to that in EV cells, which was set to 1.0. Quantification of CFP-*Nb*ATG8f-labeled **(B)** or MDC-stained **(D)** autophagic foci per cell was performed by counting autophagic bodies to calculate autophagic activity. Over 150 cells per treatment were used

for quantification. Bars indicate standard error from three individual experiments. Asterisks indicate significant differences by Student's *t*-test compared to the control (**, $p<0.01$). (TIF)

**S12 Fig. *Nb*eIF4A interferes with the interaction between *Nb*ATG5 and *Nb*ATG12. (A and B)** *Nb*ATG5 interacted with *Nb*ATG12 in a BiFC assay, while the interaction signals were reduced by *Nb*eIF4A expression (4A-myc). **(C)** Bars, 20 μm. **(C)** In a Co-IP assay, when *Nb*eIF4A was expressed, *Nb*ATG5 was immunoprecipitated, but it was difficult to immunoprecipitate *Nb*ATG12.
(TIF)

**S13 Fig. Expression of vsiRNA-4A downregulates the expression of Nb*eIF4A*.** The relative transcript level of endogenous Nb*eIF4A* was measured in cells where vsiRNA-4A was expressed via artificial miRNA expression. An unrelated small RNA (UUAAGGGUAA-GUUUUCCGCAU) expressed in the vector (Vec) was used as the control.
(TIF)

**S14 Fig. Amino acid and nucleotide sequence alignments between *Os*eIF4A and *Nb*eIF4A.** Amino acid (A) and nucleotide (B) sequence alignments.
(TIF)

**S15 Fig. Os*eIF4A* mRNA is targeted by vsiRNA-4A for cleavage. (A)** Os*eIF4A* transcript levels in RSV-infected rice at 14 dpi. **(B)** Immunoblot analysis using GFP-tagged *Os*eIF4A (*Os*eIF4A-GFP). For the control, a similar construct was prepared using a mutant *Os*eIF4A (*Os*eIF4Am-GFP) in which nucleotides 9–13 at the binding site of vsiRNA-4A were complementary to the wild type. vsiRNA-4Am represents the vsiRNA-4A mutant at nucleotides 9–13 (nucleotides in red). **(C)** GFP-tagged *Os*eIF4A accumulation when co-expressed with vsiRNA-4A and in the controls (vsiRNA-4A/*Os*eIF4Am-GFP or vsiRNA-4Am/*Os*eIF4A-GFP) at 5 dpi.
(TIF)

**S16 Fig. Amino acid sequence alignments between *Os*ATG5 and *Nb*ATG5.**
(TIF)

**S17 Fig. *Os*eIF4A suppresses *Nb*ATG5-activated autophagy. (A, B)** Representative confocal images of autophagic activity revealed by *Nb*ATG8f-CFP **(A)** or MDC staining **(B)**. **(C)** Relative autophagic activity, normalized to that of cells expressing RFP and empty vector. Quantification of autophagic foci per cell was performed by counting autophagic bodies to calculate autophagic activity. More than 150 cells per treatment were used for quantification. Error bars indicate standard error from three individual experiments. Asterisks indicate significant differences by Student's *t*-test (**, $p < 0.01$) compared to the control. **(D)** Immunoblot analysis of protein accumulation.
(TIF)

**S1 Table. Primers Used for Analysis.**
(DOCX)

## Acknowledgments

We thank Prof. Yule Liu at Tsinghua University providing all vectors used in autophagy analysis. We thank Prof. M.J. Adams, Minehead, UK for correcting the English in the manuscript.

## Author Contributions

**Conceptualization:** Jianping Chen, Fei Yan.

**Data curation:** Xiangxiang Zhang, Yueyan Yin, Yunhe Su, Zhaoxing Jia, Fei Yan.

**Formal analysis:** Xiangxiang Zhang, Yueyan Yin, Yunhe Su, Liangliang Jiang, Yuwen Lu, Hongying Zheng, Jiejun Peng, Shaofei Rao, Guanwei Wu, Jianping Chen, Fei Yan.

**Funding acquisition:** Shaofei Rao, Fei Yan.

**Methodology:** Xiangxiang Zhang, Zhaoxing Jia, Liangliang Jiang.

**Project administration:** Jianping Chen, Fei Yan.

**Supervision:** Fei Yan.

**Writing – original draft:** Xiangxiang Zhang, Fei Yan.

**Writing – review & editing:** Jianping Chen, Fei Yan.

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
