## [Decision Letter · Decision Letter 0]

16 Aug 2021

Dear Dr. Yan,

Thank you very much for submitting your manuscript "eIF4A, a target of siRNA derived from rice stripe virus, negatively regulates antiviral autophagy by interacting with ATG5 in Nicotiana benthamiana" for consideration at PLOS Pathogens. As with all papers reviewed by the journal, your manuscript was reviewed by members of the editorial board and by several independent reviewers. The reviewers appreciated the attention to an important topic. Based on the reviews, we are likely to accept this manuscript for publication, providing that you modify the manuscript according to the review recommendations.

Sincerely,

Savithramma P. Dinesh-Kumar

Associate Editor

PLOS Pathogens

Peter Nagy

Section Editor

PLOS Pathogens

Kasturi Haldar

Editor-in-Chief

PLOS Pathogens

orcid.org/0000-0001-5065-158X

Michael Malim

Editor-in-Chief

PLOS Pathogens

orcid.org/0000-0002-7699-2064

Reviewer Comments (if any, and for reference):

Reviewer's Responses to Questions

**Part I - Summary**

Reviewer #1: Authors have carefully addressed most of the issues raised in the last review. I have just a couple of comments.

Reviewer #2: The authors well revised the manuscript by addressing issues that the editor and reviewers raised with additional or re-examined data. Although not wholly convinced yet with complex interactions among RSV, host RNA silencing, and autophagy described in this study, this reviewer agrees that this study is worth being published in terms of its suggesting multiple or concatenated interactions of viral vsiRNAs, silencing suppressor, host RNA silencing and autophagy. The revised still include inappropriate descriptions and errors listed below.

**Part II – Major Issues: Key Experiments Required for Acceptance**

Reviewer #1: The use of Arabidopsis ATG12 in Fig. S12. To reveal the role of NbeIF4A in regulation of autophagy, NbATG12 is more appropriate than AtATG12.

Reviewer #2: 1. Fig. S2, no information (nucleotide sequence) of primers for RT-PCR to detect RSV was provided.

2. L137-141, the authors described "Several defense pathways participate in plant responses to viruses, including RNA silencing, phytohormone signaling, autophagy, and 26S proteasome-mediated

139 degradationTo determine whether these pathways help prevent RSV infection in NbeIF4A-silenced plants, we investigated the expression of marker genes for each pathway upon NbeIF4A silencing." However, The authors did not provide the results, except for those related to autophagy in Fig. S3. Provide the other results or rewrite these sentences.

3. Fig. 3 and Fig. S6, Describe how the TuMV CP, RSV CP and P3 proteins were detected. If specific antibodies were used, did the authors buy them or prepare by themselves?

**Part III – Minor Issues: Editorial and Data Presentation Modifications**

Reviewer #1: 1. Page 6, lines 137-144: The text implies that the authors measured the transcript levels of RNAi and other genes related to antiviral defense. However, I could not find the data relevant to this point. Please provide the data or revise the text appropriately.

2. Page 6, line 162: Provide a reference for the role of NbeIF6A in translation.

3. Fig. S4: The authors used a 35S-driven GFP construct as a proxy for protein translation. The result supports the authors’ claim but does not rule out that NbeIF4A has any role in RSV translation.

4. Fig. 3A-C: The experimental design for this experiment is missing in the main text. The authors might use 16c Nb plants to address the effect of NbeIF4A-silencing on the VSR activity of the RSV p3 protein. It would be helpful if the authors could briefly describe this point in the main text (Page 8).

5. Page 9, lines 236-238: To be more accurate, the authors should provide all the repeats relevant to Fig. 3F and include statistical analysis of accumulation level of RSV CP in the supplement.

Reviewer #2: L 31, rice stripe virus (RSV) (all lowercase letters and roman)

L 78, "Autophagy-related genes (ATGs) are key factors" should be "Autophagy-related genes (ATGs) encode key factors"

L 104, "encoding eukaryotic translation initiation factor 4A (eIF4A)" should be "encoding eIF4A"

L 182, Is the description: "monitored the progression of infection with the CFP-NbATG8f reporter", OK? with GFP

L 187, "the TuMV-GFP chimeric virus" should be "GFP-tagged TuMV"

Regarding the TuMV-GFP, describe how the authors got the infectious clone and provide the reference regarding the infectious clone of GFP-tagged TuMV.

All "Nib" in the text should be "NIb"

L 196-197, Fig. S6C just investigated the accumulation of NIb-GFP and the authors did not provide evidence proving the degradation of NIb-GFP. Rewrite the sentence.

PLOS authors have the option to publish the peer review history of their article (what does this mean?). If published, this will include your full peer review and any attached files.

Reviewer #1: No

Reviewer #2: No

Figure Files:

Data Requirements:

Reproducibility:

References:

---

## [Editor Report · Decision Letter 1]

21 Sep 2021

Dear Dr. Yan,

We are pleased to inform you that your manuscript 'eIF4A, a target of siRNA derived from rice stripe virus, negatively regulates antiviral autophagy by interacting with ATG5 in Nicotiana benthamiana' has been provisionally accepted for publication in PLOS Pathogens.

Best regards,

Savithramma P. Dinesh-Kumar

Associate Editor

PLOS Pathogens

Peter Nagy

Section Editor

PLOS Pathogens

Kasturi Haldar

Editor-in-Chief

PLOS Pathogens

orcid.org/0000-0001-5065-158X

Michael Malim

Editor-in-Chief

PLOS Pathogens

orcid.org/0000-0002-7699-2064
---

## [Editor Report · Acceptance letter]

24 Sep 2021

Dear Dr. Yan,

We are delighted to inform you that your manuscript, "eIF4A, a target of siRNA derived from rice stripe virus, negatively regulates antiviral autophagy by interacting with ATG5 in Nicotiana benthamiana," has been formally accepted for publication in PLOS Pathogens.

Best regards,

Kasturi Haldar

Editor-in-Chief

PLOS Pathogens

orcid.org/0000-0001-5065-158X

Michael Malim

Editor-in-Chief

PLOS Pathogens

orcid.org/0000-0002-7699-2064